# Peer review of "Relative Validity of the Eat and Track (EaT) Smartphone App for Collection of Dietary Intake Data in 18-to-30-Year Olds"

_nutrients, 2019, doi:10.3390/nu11030621_

Round 1

Reviewer 1 Report

Thank you for the opportunity to read and review this paper investigating the validity of a smartphone app for dietary intake measurement compared to 24h recall. I think the paper is important and of interest to the readers. However, I feel the paper would benefit from major revisions. First of all, I think the role of eating out or eating fast food as the strength of the study should be clarified. Second, I encourage the authors to be more modest and realistic in their conclusions. Although I feel that the app could be useful (at least in some populations), there are considerations, which have to be taken into account when using the app. Please see below for my detailed comments.

Abstract

Point 1: Line 18: What does ‘higher socio-economic status’ mean here? Did you measure income or something else? I suggest changing the term ‘socio-economic status’ to something that describes the measured covariate more accurately.

Point 2: Line 20: Differences in mean energy intake?

Introduction

Point 3: I find the introduction a bit misleading. I think it would benefit from reordering: maybe consider starting from smartphone apps for dietary intake measurement and later proceed to explaining the strengths of your particular app (fast food data base)?

Point 4: Lines 47-48: Which nutrients have been reported in these previous studies? Has the agreement been poor for all of the nutrients or just some of them? Why do you think they did not have good agreement?

Point 5: Line 57: You name 24h recall as the ‘gold standard’. However, later in the discussion you acknowledge that it is not actually the ‘gold standard’. Maybe you could just write that the validation studies have used 24h recalls as reference method?

Point 6: Line 59: The target age group here refers to your study’s target age group? I suggest adding the age group just in case the reader does not remember that.

Point 7: Lines 62-70: I suggest removing this paragraph from the introduction and adding it to the methods section.

Point 8: Just to clarify: is eating out here the same thing as eating fast food? Is fast food everything from sushi and ready-made salads to burgers and fries?

Materials and methods

Point 9: How many potential participants completed the screening? How many participants there were in the MYMeals study?

Point 10: Line 82: I suppose NSW means New South Wales? I suggest not using the abbreviation since not everyone might be familiar with it.

Point 11: Line 83: Were the MYMeals participants also 18-30-year-olds? Were these inclusion criteria specific for the validation study or were they inclusion criteria for the MYMeals study? Please state clearly, which of the criteria were specific to the validation study and which were not.

Point 12: Line 87-88: Is this the same thing that was already reported in the lines 76-77?

Is the app available in iTunes or Google Play? I would like to have a more detailed description of the development of the app (who developed it, for what purposes (research or commercial), can anyone use it etc.).

Point 13: Lines 106-107: Were the days for recording the foods preset by the research group or did the participants choose their own days?

Point 14: I would like to have a more elaborate description of the 24h recall procedure. For example, did the dietitians enter the data at the same time as they interviewed the participants? Which timeframes did the interview cover? Did the timeframe match that of the EaT app? Did you check the data? How?

Point 15: Why did you not use the same database with the same fast food items in the 24h recall?

Do you think that the gift voucher could somehow affect the results? Was the value of the voucher 100 USD or 100 AUD?

Point 16: Line 121: When where the EaT app entries checked by dietitians? At the same time as the 24h recall was completed or was there another contact? How long after the entry was made?

Point 17: Line 122: Which were the obvious errors that prompted the contact? How did you decide that a unit size was incorrect or a meal was skipped? How many participants were contacted?

Point 18: Line 124: Who did the matching of the manually entered foods? How many such entries there were in the data?

Point 19: Line 131: “Individual means were calculated across the three days…” Do you mean you calculated means for for example energy intake in three days or did you calculate mean daily intakes?

Point 20: How did you choose the nutrients to compare? Is there any special reason you chose to also compare sodium intake (besides macronutrients) measured by the two methods? What kind of instruction did you give for the participants with regard to salt? Were they instructed to pay extra attention to salt? Do you think they can reliably report the salt contents of everything they ate?

Point 21: Line 150: Did I understand correctly that you did not exclude the 16 under-reporters? Why was that? How would that affect your results?

Results

Point 22: Lines 154-156: Did the 24 drop-outs differ from the participants who were included in the analyses? Do the numbers match? 216 participants were recruited, 5 withdrew and 19 did not complete the records à 192 participants were in the final sample?

Point 23: Table 1: The table could be easier to read, if there were some horizontal lines between the different measures. Please elaborate what measures were included in the socio-economic index of the area. Similarly, the educational levels might need some more clarification (for example, what does “trade or diploma” mean?). Please add the BMI cut points for the categories.

Point 24: Table 2: Are the numbers in the two middle columns means or medians? And the numbers in the parenthesis, are they IQRs? Are there any means and SDs in the table? Please make the column headings as accurate as possible.

Point 25: The Figures are not clear enough to read properly.

Discussion

Point 26: Lines 229-230: I’m sorry if I didn’t get it, but you didn’t use an energy-adjustment in the current paper, right? Maybe consider rewording the sentence and making it clearer in which study you used energy-adjustment.

Point 27: It would be nice to read your thoughts regarding gender differences in the correlation coefficients. Why do you think that females had lower correlations and proportions of participants classified into the same quarter?

Point 28: In the paper, you did not report how often the participants ate fast food/ate something outside of the home, but you still state your fast food item including database as your study’s strength. It would be nice to know how frequent that was.

Point 29: Lines 265-266: Why does the EaT app clear its history at 3am? Would it be good to have an option to add something on the next day too?

Point 30: For which kinds of studies and populations you think the EaT app is a promising method? On which level you think that it can be used to measure food intake (total diet/food group/food/macronutrient/micronutrient)? Which levels you wouldn’t feel safe to use?

Point 31: Do you think that, compared to traditional pen&paper food record, which is often referred to as the ‘gold standard’, the app is more burdensome for the participants or researchers? Why? Whose burden we should aim to minimize?

Point 32: Lines 268-270: I do not understand this sentence, please reword.

Conclusions

Point 33: The conclusions are maybe a bit too positive. I would add something about the level on which the app is valid enough to be used.

Author Response

Thank you for the opportunity to read and review this paper investigating the validity of a smartphone app for dietary intake measurement compared to 24h recall. I think the paper is important and of interest to the readers. However, I feel the paper would benefit from major revisions. First of all, I think the role of eating out or eating fast food as the strength of the study should be clarified. Second, I encourage the authors to be more modest and realistic in their conclusions. Although I feel that the app could be useful (at least in some populations), there are considerations, which have to be taken into account when using the app. Please see below for my detailed comments.

Thank you for these comments.

Abstract

Point 1: Line 18: What does ‘higher socio-economic status’ mean here? Did you measure income or something else? I suggest changing the term ‘socio-economic status’ to something that describes the measured covariate more accurately.

Higher socio-economic status is determined by the Australian Bureau of Statistics’ Socio-economic Indexes for Areas based on the postcode of the participant’s residence. This is a measure of the socio-economic status of the area that the participant lives in. The Bureau uses the national Census data on income, education, economic status, advantage and disadvantage. To explain this, we have added the words ‘60% resided in higher socio-economic areas’ at line 19-20.

Point 2: Line 20: Differences in mean energy intake?

That is correct and has been added at line 21.

Introduction

Point 3: I find the introduction a bit misleading. I think it would benefit from reordering: maybe consider starting from smartphone apps for dietary intake measurement and later proceed to explaining the strengths of your particular app (fast food data base)?

We have considered the comments of the other three reviewers who did not request this. However, they did ask for a section of introduction to be moved into methods which we have done and we hope this makes the introduction better now.

Point 4: Lines 47-48: Which nutrients have been reported in these previous studies? Has the agreement been poor for all of the nutrients or just some of them? Why do you think they did not have good agreement?

Lines 48-52 have been revised, and now read:

“A number of commercial diet collection apps are available, however, validation studies show that these do not have good agreement with established dietary methods for some nutrients, including energy, protein, total fat, sugars, fibre and sodium [10,11]. Such differences may be due to the underlying nutrition composition databases, inadequacy of food listings available and no accounting for food preparation methods.” 

Point 5: Line 57: You name 24h recall as the ‘gold standard’. However, later in the discussion you acknowledge that it is not actually the ‘gold standard’. Maybe you could just write that the validation studies have used 24h recalls as reference method?

This has now been changed to reference as suggested, see line 61.

Point 6: Line 59: The target age group here refers to your study’s target age group? I suggest adding the age group just in case the reader does not remember that.

We have changed ‘the target age group’ to ‘young adults’ at line 64.

Point 7: Lines 62-70: I suggest removing this paragraph from the introduction and adding it to the methods section.

Lines 62-70 have been moved to the Methods.

Point 8: Just to clarify: is eating out here the same thing as eating fast food? Is fast food everything from sushi and ready-made salads to burgers and fries?

Fast foods are those sourced from chain outlets with standardised menus. Eating out can be from chain outlets, but also encompasses independent outlets, such as local cafes, restaurants and takeaways.

Materials and methods

Point 9: How many potential participants completed the screening? How many participants there were in the MYMeals study?

In total, there were 2903 attempts of the screening survey. Of these, 287 did not complete the screening survey, 89 were screened out, 69 were duplicates (participants trying to complete the study multiple times) and 6 were test entries by the study team. This left 2452 eligible entries. Of these, 107 withdrew (including 5 from the validation study as noted in line 170), 339 had full quotas before they signed up and 962 were lost to follow up after completing the screening. The total completions were 1044. This information will be included in the main MYMeals study publication so to avoid future self-plagiarism we will not include this data here rather just the numbers and drop outs from the validation study.

Point 10: Line 82: I suppose NSW means New South Wales? I suggest not using the abbreviation since not everyone might be familiar with it.

NSW has been expanded to ‘New South Wales’.

Point 11: Line 83: Were the MYMeals participants also 18-30-year-olds? Were these inclusion criteria specific for the validation study or were they inclusion criteria for the MYMeals study? Please state clearly, which of the criteria were specific to the validation study and which were not.

The criteria were the same for both the overarching MYMeals Study and the validation sub-study. This has been clarified at line 75-84.

Point 12: Line 87-88: Is this the same thing that was already reported in the lines 76-77?

Lines 76-77 is about what was collected, lines 87-88 is about how.

Is the app available in iTunes or Google Play? I would like to have a more detailed description of the development of the app (who developed it, for what purposes (research or commercial), can anyone use it etc.).

As stated in Line 115-117  “After obtaining consent, participants were emailed links to download the EaT app from either the Apple App Store or Google Play…”  Although the app can be downloaded from the store by the general public, they will not be able to access the data entered nor any nutrition information as it is currently a data collection app for research purposes only.

The app was created by the researchers working with our colleagues in the School of Information Technology. This has been described in more detail in another publication by our group reference 18. A detailed paper on the development of the EaT app is referenced in another paper (reference 18 – previously 19).

Point 13: Lines 106-107: Were the days for recording the foods preset by the research group or did the participants choose their own days?

The participants were instructed by the researchers as to which days they should record their food and beverage intakes. This is stated in the Methods, “The researchers instructed participants on the days they must record their intakes. The starting days were staggered across the population to facilitate an even spread of days over the week.” At lines 118-120

Point 14: I would like to have a more elaborate description of the 24h recall procedure. For example, did the dietitians enter the data at the same time as they interviewed the participants? Which timeframes did the interview cover? Did the timeframe match that of the EaT app? Did you check the data? How?

As stated in lines 122-3, the automated, online ASA-24 Australia system was used by the research dietitians at the time of the interview (see also references 9 and 23). We have added more detail to lines 126-130:

“The participants also completed three 24-hour recall telephone interviews with research dietitians. To allow all foods to be captured by both methods, the 24-hour recalls were conducted the following day, but captured data for the same days that the app was used. The automated, online ASA-24 Australia [9,25] was used to conduct the recalls so that the interview process was standardised. This computerised method involved the dietitian recording all foods and drinks consumed throughout the day into the ASA-24 Australia as they interviewed the participants. Multiple passes prompt for additional information on food form, preparation methods, portion size and omitted items. The three 24-hour recalls were conducted on the days following each of the data collection days. The ASA-24 Australia uses the AUSNUT 2011-13 database [9,26], but differs from the EaT app with respect to the number of fast food items available [18].”

We have added further information at line146.

All entries for the ASA-24 Australia recall were downloaded and checked.

Point 15: Why did you not use the same database with the same fast food items in the 24h recall?

The advantage of the ASA-24 system is the standardisation offered in the recall protocol with simultaneous calculation of all nutrients that the researcher is able to retrieve at the end of the recall as well as the entire record of food entries. This bypasses the need to re-enter any data into a food analysis program. The ASA-24 Australia has the same database of foods as the EaT app but with generic options for the fast foods rather than the complete listing of the EaT app. However, the dietitians doing the interviews have the capability to select the appropriate option in the ASA-24 recall that participants do not.

Do you think that the gift voucher could somehow affect the results? Was the value of the voucher 100 USD or 100 AUD?

The voucher was $100AUD (added to line 134). We are unable to speculate as to the effects of the voucher and to do so would not constitute a scientifically valid conclusion.

Point 16: Line 121: When where the EaT app entries checked by dietitians? At the same time as the 24h recall was completed or was there another contact? How long after the entry was made?

The entries were checked in the week following the 3 data collection days which were after the recalls. This has been added to line 137.

Point 17: Line 122: Which were the obvious errors that prompted the contact? How did you decide that a unit size was incorrect or a meal was skipped? How many participants were contacted?

Unit size such as consuming 250 serves or 10000g of a food were obvious errors. If no entry is offered for breakfast, lunch, dinner or snacks then there is a likelihood of a skipped meal. The app asks participants to record foods under these headings so they are readily identified. In total, 111 participants were contacted, but that included clarifying the manually added foods, along with checking for skipped meals and errors.

Point 18: Line 124: Who did the matching of the manually entered foods? How many such entries there were in the data?

This was conducted by one author, who is an Accredited Practising Dietitian, then checked by two others. There were 33 manually entered foods out of over 7700. This has been entered at lines 141-142.

Point 19: Line 131: “Individual means were calculated across the three days…” Do you mean you calculated means for for example energy intake in three days or did you calculate mean daily intakes?

Please see revision at line 148-149.

Daily totals for energy and each nutrient were summed, then means were calculated for each participant for the three study days

Point 20: How did you choose the nutrients to compare? Is there any special reason you chose to also compare sodium intake (besides macronutrients) measured by the two methods? What kind of instruction did you give for the participants with regard to salt? Were they instructed to pay extra attention to salt? Do you think they can reliably report the salt contents of everything they ate?

The nutrients selected are widely recognised as nutrients of public health concern, as excess consumption is associated with chronic disease. These are also nutrients that are required on nutrition information panels in Australia, and therefore these data are routinely provided by fast food chains.

Participants were not instructed to pay any specific attention to any food or nutrient. They were asked to enter everything they ate or drank, and select entries for each food from the app. We did not ask them to record table salt as our interest is the sodium provided by the foods themselves as foods prepared outside the home were hypothesized to contain higher sodium levels.

Point 21: Line 150: Did I understand correctly that you did not exclude the 16 under-reporters? Why was that? How would that affect your results?

Excluding the under-reporters did not alter the results. This has been included at line 167-168. ‘The full sample was used for analysis as removing mis-reporters did not significantly change results.’

Increasingly it is suggested to do analysis on the total sample and then test after removal of under-reporters as we have done here.

Results

Point 22: Lines 154-156: Did the 24 drop-outs differ from the participants who were included in the analyses? Do the numbers match? 216 participants were recruited, 5 withdrew and 19 did not complete the records à 192 participants were in the final sample?

Apologies, we have reanalysed and corrected numbers for a total 189.

Point 23: Table 1: The table could be easier to read, if there were some horizontal lines between the different measures. Please elaborate what measures were included in the socio-economic index of the area. Similarly, the educational levels might need some more clarification (for example, what does “trade or diploma” mean?). Please add the BMI cut points for the categories.

Table 1 is formatted as per the Journal’s guidelines.

The categories we have used here for demographics are those that the Australian Government employs. We wish to be consistent with the Australian Bureau of Statistics classifications of the Australian population. This is important if one wants to consider generalisability of a study. As stated previously and referenced in the footnote, socio-economic status is determined by the Australian Bureau of Statistics who has assigned each residential postcode area in the country a Socio-economic Index for Area. This assesses the relative disadvantage in an area based on data collected in the national census. Each postcode in Australia has had the education, income, economic situation and other measures of advantage and disadvantage measured and is then assigned a score.

Trade or diploma are the official classifications for post-school qualifications that are not University degrees.

BMI cut-offs have been added to Table 1.

Point 24: Table 2: Are the numbers in the two middle columns means or medians? And the numbers in the parenthesis, are they IQRs? Are there any means and SDs in the table? Please make the column headings as accurate as possible.

The column headings are correct. The two middle columns are medians and interquartile ranges for energy and nutrient densities for consistency. The superscript footnote references in the left hand column explain the statistical tests that were used.

Point 25: The Figures are not clear enough to read properly.

The figures have been reinserted so they are more legible.

Discussion

Point 26: Lines 229-230: I’m sorry if I didn’t get it, but you didn’t use an energy-adjustment in the current paper, right? Maybe consider rewording the sentence and making it clearer in which study you used energy-adjustment.

Yes, we used energy adjustment (macronutrients expressed as percentage of total energy intake, and sodium as an energy density) as qualified in this sentence.

Point 27: It would be nice to read your thoughts regarding gender differences in the correlation coefficients. Why do you think that females had lower correlations and proportions of participants classified into the same quarter?

Correlation is a measure dependent on the spread of intakes. Most nutrition epidemiologists now consider the use other measures such as Bland Altman for validation to be better thus we prefer not to add additional discourse on the correlation coefficients. You will see we reanalysed all data and differences not obvious.

Point 28: In the paper, you did not report how often the participants ate fast food/ate something outside of the home, but you still state your fast food item including database as your study’s strength. It would be nice to know how frequent that was.

Yes, this is very interesting, but this is the focus of our main manuscript for the MYMeals Study not for the validation of the dietary intake app. So we prefer to keep these findings unpublished until then.

Point 29: Lines 265-266: Why does the EaT app clear its history at 3am? Would it be good to have an option to add something on the next day too?

There are both technological and validity issues here. The whole aim of diet recording (as opposed to recall methods such as 24hr recall is that it should be prospective with recording as participants consume the food, and this is how participants were instructed. Keep in mind that if participants had access to their previous day’s data it would compromise the validity of using the 24hr recall as an assessment method. What might be desired in an app for managing one’s diet with progressive records is quite different to what is required for research purposes assessing dietary intakes.

Point 30: For which kinds of studies and populations you think the EaT app is a promising method? On which level you think that it can be used to measure food intake (total diet/food group/food/macronutrient/micronutrient)? Which levels you wouldn’t feel safe to use?

The concluding paragraph (lines 303-305) outlines where we think the EaT app is promising:

‘Overall, EaT is a promising method of collecting dietary intake data of young adults, with a particular focus on eating out. The EaT app could be used to collect data investigating the types and contributions of nutrients from different types of food outlets, and investigate effects of environmental interventions in fast food chain outlets.’

As we have not validated the EaT app for food groups or nutrients other than those included in the study, we prefer not to speculate on these as they would not be scientific recommendations.  

Point 31: Do you think that, compared to traditional pen&paper food record, which is often referred to as the ‘gold standard’, the app is more burdensome for the participants or researchers? Why? Whose burden we should aim to minimize?

This is not the focus of this paper. We do not aim to give a general overview of pros and cons of apps. There are prior reviews including by some of the authors of the current paper. Previous research in the Australia and UK has indicated that participants prefer apps for recording than pen and paper because it is convenient to always carry a smartphone with them. The burden of both is reduced but obviously more for the researcher.

Point 32: Lines 268-270: I do not understand this sentence, please reword.

We do not understand how the sentence does not make sense. We are open to editorial suggestion.

Conclusions

Point 33: The conclusions are maybe a bit too positive. I would add something about the level on which the app is valid enough to be used.

Thank you we have now ‘softened’ this conclusion:

“The Eat and Track smartphone application is a valid way of collecting group nutrient density intake data in 18 to 30 year olds, with a specific focus on the nutrients of interest when frequently eating out i.e. sugars, saturated fat and sodium. To further assess the validity of the app, additional methods that do not rely on food and beverage capture and nutrient databases, such as biomarker or doubly-labelled water studies should be conducted.”

Reviewer 2 Report

To the authors,

Comments and suggestions:

Title: the term relative validity would be a better description of this work than comparative validation

line 82, for an international audience please explain/describe “NSW”

In line ine98 you describe the portion booklet. This is not mentioned anywhere else. Please add to the discussion how the participants used the portion measure booklet and if you know to what extent it was used actively by the participants in estimating portion sizes.  Also please discuss how the method would work without this booklet.

Nutrient density identifies the proportion of nutrients in foods. In line 72 you say that you will examine energy and nutrient densities. I think you mean nutrient intakes. In the results section you present energy intake end energy percent, and one value for salt /energy intake. Please rephrase line 72, and also throughout the manuscript, so that it matches your results.

Line 111 “….for days that matched with those of the Eat app.” What does this mean? Matched how? Further down in the text you describe that the interviews were done on the following day of the registration. Is that what you mean by matched? Please rephrase and clarify.

In the method section please add a paragraph about the food composition databases used. In particular describe the food composition data in the “fast food database” and how it was compiled. This is relevant for your results which only present evaluations of the energy providing nutrients.

I assume that the intake unit of energy should be given in kJ/day?

All figures of Bland-Altman plots are in the present format not readable.

Line 225. You did find significant differences in energy, carbohydrates and sugar intakes between the 2 methods. Please rephrase line 225 based on this.

Line 226. There was a slight bias in the Bland-Altman plot for carbohydrates in male, figure 2 c. Please nuance your statement.

Line 246- please add information on the general population percentages in the manuscript so that the reader may compare and understand your conclusion on the representativeness of your population.

Line 260: “…the fast foods in the EaT app had greater nutrient ranges and higher maximum values.” This statement stands alone with no explanation or data to back it. There is no information on the food composition database used. And the validation study presented in the manuscript is only focused on a small number of energy providing nutrients and salt.

Conclusions. The conclusion that the application is a valid way of collecting data is only evaluated for a small number of nutrients. Please rephrase and specify.

Author Response

Title: the term relative validity would be a better description of this work than comparative validation

The title has been updated to state ‘relative validity’.

line 82, for an international audience please explain/describe “NSW”

NSW has been expanded to ‘New South Wales’. The description is following the comma.

In line 98 you describe the portion booklet. This is not mentioned anywhere else. Please add to the discussion how the participants used the portion measure booklet and if you know to what extent it was used actively by the participants in estimating portion sizes.  Also please discuss how the method would work without this booklet.

‘For other foods participants received a portions booklet used in national nutrition surveys to estimate serving sizes but moving forward the inclusion of images within the app may be advantageous’ has now been added to the Discussion at line 284-285.  We did not conduct a process evaluation so our ability to assess how frequently the portion booklet was used is limited and not conclusive of usage.

Nutrient density identifies the proportion of nutrients in foods. In line 72 you say that you will examine energy and nutrient densities. I think you mean nutrient intakes. In the results section you present energy intake and energy percent, and one value for salt /energy intake. Please rephrase line 72, and also throughout the manuscript, so that it matches your results.

The aim of this work was to examine the percentage energy provided by different nutrients. Percentage energy from a nutrient i.e kJ from the nutrient per 1000kJ is nutrient density of the diet. Sodium was presented per 1000kJ. If we were to say nutrient intakes we would be providing nutrients as gram intakes. We have deliberately not presented data in this way because nutrient densities may correlate more closely with true intakes than absolute intakes. See this article for reference: Park Y, Dodd KW, Kipnis V,et al. Comparison of self-reported dietary intakes from the Automated Self-Administered 24-h recall, 4-d food records, and food-frequency questionnaires against recovery biomarkers. Am J Clin Nutr. 2018 Jan 1;107(1):80-93

Line 111 “….for days that matched with those of the Eat app.” What does this mean? Matched how? Further down in the text you describe that the interviews were done on the following day of the registration. Is that what you mean by matched? Please rephrase and clarify.

They were the same days that they were recording with the EaT app. The interviews are done on the following day so that you can capture everything that was eaten the day before, otherwise it would be collected prospectively and not what was actually consumed. Lines 122-126 now read:

‘The participants also completed three 24-hour recall telephone interviews with research dietitians. To allow all foods to be captured by both methods, the 24-hour recalls were conducted the following day, but captured data for the same days that the app was used.’

In the method section please add a paragraph about the food composition databases used. In particular describe the food composition data in the “fast food database” and how it was compiled. This is relevant for your results which only present evaluations of the energy providing nutrients.

This has now been added at lines 99-101:

“The fast food nutritional composition data is currently restricted to energy, protein, total and saturated fats, carbohydrates, sugars, and sodium and does not contain micronutrients.” It is described in detail in reference 18.

I assume that the intake unit of energy should be given in kJ/day?

Correct

All figures of Bland-Altman plots are in the present format not readable.

The figures have been reinserted so they are more legible.

Line 225. You did find significant differences in energy, carbohydrates and sugar intakes between the 2 methods. Please rephrase line 225 based on this.

This has been rephrased now

“This finding is based on non-significant differences in group intakes with the exception of carbohydrates,…”

Note we reanalysed all the data and carbohydrate and energy are the only differences.

Line 226. There was a slight bias in the Bland-Altman plot for carbohydrates in male, figure 2 c. Please nuance your statement.

You are correct and this now reads line 213-214 “For males carbohydrate showed underestimation at lower intakes and overestimation at higher intakes with the app compared with 24-h recalls.”

Line 246- please add information on the general population percentages in the manuscript so that the reader may compare and understand your conclusion on the representativeness of your population.

A comment on this has been added to the results section. Line 176-178

‘It should be noted there are slightly fewer males in the study than the Australian population proportion of 49%; and the proportion with post-school qualifications (65%) is more than the 56% reported by the Australian Bureau of Statistics [32].’

The socioeconomic status was dichotomised into high and low SES according to the Australian data and should be 50:50 so slightly higher SES in our sample (60%).

Line 260: “…the fast foods in the EaT app had greater nutrient ranges and higher maximum values.” This statement stands alone with no explanation or data to back it. There is no information on the food composition database used. And the validation study presented in the manuscript is only focused on a small number of energy providing nutrients and salt.

Yes. As we have explained, the validation is for the nutrients that are of most interest when it comes to consumption of foods prepared outside the home. We have extensively validated our e-DIA for other research studies. The food composition database has previously been the subject of two research papers that have already been published (references 18 and 22). We have referenced these again at line 292.

Conclusions. The conclusion that the application is a valid way of collecting data is only evaluated for a small number of nutrients. Please rephrase and specify.

This has now been rephrased:

‘…with a specific focus on the nutrients of interest when frequently eating out i.e. sugars, saturated fat and sodium.’

Reviewer 3 Report

The development of new and innovative methods of measuring diet is important to our ongoing understanding of diet and health links. This paper examines the relative validity of a new smart phone app against the 24 hour recall method. The methods are well described and appropriate conclusions drawn from the research.

Methods

How was the sample size of 20% of the My Meals study decided?

Discussion

An aim of the study was to include participants more reflective of Australian young adults with respect to education and socio-economic status. This is noted as a strength of the study.  It would be useful to show how the sample compares the Australian general population for this age group to show in fact it achieved this goal. 48% had a University degree- how does this compare.

A limitation of the study discussed by the authors is that completing the EaT app before the 24 hr recall could have improved the accuracy of the recall. Do the authors have any insight into whether using the EaT app could in itself changed diet?

Author Response

The development of new and innovative methods of measuring diet is important to our ongoing understanding of diet and health links. This paper examines the relative validity of a new smart phone app against the 24 hour recall method. The methods are well described and appropriate conclusions drawn from the research.

Thank you for these comments.

Methods

How was the sample size of 20% of the My Meals study decided?

In our successful grant refereed by the Australian Research Council we proposed 20% of participants for the validation study. It is widely acknowledge that a minimum 100 is included in a validation study. We increased our sample size above this to account for drop outs and to ensure a wider representation of participants with various demographic characteristics would be present in the sample to improve generalisability.

Discussion

An aim of the study was to include participants more reflective of Australian young adults with respect to education and socio-economic status. This is noted as a strength of the study.  It would be useful to show how the sample compares the Australian general population for this age group to show in fact it achieved this goal. 48% had a University degree- how does this compare.

We have added information into the results sections so that the comparisons are more readily made. The 2016 Census of Population and Housing has recorded that 56% of Australians aged 15 years and over – 9.6 million people – hold a post-school qualification.

A limitation of the study discussed by the authors is that completing the EaT app before the 24 hr recall could have improved the accuracy of the recall. Do the authors have any insight into whether using the EaT app could in itself changed diet?

Participants were instructed to enter what they ate and drank, and not to change how they ate. But we have no record of their intakes before the study so we do not have any insight into this and we prefer not to speculate. One advantage of this app is that it provides no feedback on energy or nutrient intakes that may introduce a bias and the entries are cleared each day at 3:00am.

Reviewer 4 Report

The paper entitled “Comparative validity of the Eat and Track (EaT) smartphone app for collection of dietary intake data in 18 to 30 year olds” addressed a validation of a novel method of dietary assessment; a very interesting issue for Nutrients’ readers. This is a well-written work although there are some concerns about several issues that the authors should carefully tackle before considering the manuscript for publication.

1) To validate the EaT app, the authors used 24-hour recall telephone interviews although they declared some differences between methods regarding the number of fast food items. Considering that mentioned in the Introduction, the main interest should be focused on the fast food items. In this sense, it was interesting that the authors mentioned something about these items, taking into account that they could be a likely explanation of the differences observed.

2) The authors mentioned that they used the full sample due to only 8% of the participants were classified as under-reported by the reference 24-hour recall data. Did the authors check is there were differences between two data sets? It could be an important bias in the study?

3) The validation of the Eat app is based on the examination of energy and nutrient densities but why do the authors not mentioned anything about the foods or types of foods?

4) The authors reported a poor agreement for energy intake although they declared that they applied an energy-adjustment in their study. Considering the measurement errors in the assessment of nutrient intakes are strongly correlated with errors in the measurement of total energy intake, it is recommended that the authors also applied the well-known residual method by Walter Wilet.

Minor issues

The figures presented in the manuscript have a very low quality.

Author Response

The paper entitled “Comparative validity of the Eat and Track (EaT) smartphone app for collection of dietary intake data in 18 to 30 year olds” addressed a validation of a novel method of dietary assessment; a very interesting issue for Nutrients’ readers. This is a well-written work although there are some concerns about several issues that the authors should carefully tackle before considering the manuscript for publication.

Thank you for these comments.

1) To validate the EaT app, the authors used 24-hour recall telephone interviews although they declared some differences between methods regarding the number of fast food items. Considering that mentioned in the Introduction, the main interest should be focused on the fast food items. In this sense, it was interesting that the authors mentioned something about these items, taking into account that they could be a likely explanation of the differences observed.

We do not completely understand what question the reviewer is asking. We have provided further information about the recall. The thrust of our paper is to determine the relative validity of the EaT app for macronutrient densities. Food prepared outside the home is of concern because of the high saturated fat, sugars and sodium contents.

2) The authors mentioned that they used the full sample due to only 8% of the participants were classified as under-reported by the reference 24-hour recall data. Did the authors check is there were differences between two data sets? It could be an important bias in the study?

First we apologise but the under-reporting is 14.8% not 8%. The reason we include the entire data set is because removal of under or over-reporters on the basis of the reference method did not change the significance of the findings. This has been included at line 168 : ‘The full sample was used for analysis as removing mis-reporters did not significantly change results.’

We have misreporting in the app but we do not want to remove under-reporters to make our results look better. This is to show close to the true validity of the app.

3) The validation of the Eat app is based on the examination of energy and nutrient densities but why do the authors not mentioned anything about the foods or types of foods?

As stated in reply 1 the aim of the study was to assess these nutrient densities. We believe this provides a useful addition to the literature and in keeping with the nutrient-based validations of Griffiths 2018 (reference 11), Chen 2018 (reference 12), Rangan 2015 (reference 14), Ambrosini 2018 (reference 16) Pendergast 2017 (reference 31).

4) The authors reported a poor agreement for energy intake although they declared that they applied an energy-adjustment in their study. Considering the measurement errors in the assessment of nutrient intakes are strongly correlated with errors in the measurement of total energy intake, it is recommended that the authors also applied the well-known residual method by Walter Wilet.

We chose to use nutrient densities to adjust for energy instead of using the residual method. As per Subar (2015, reference 33), nutrients reported as percentage energy is a recommended method of adjusting nutrients for energy and assessing nutrient intakes in dietary intake data.

Minor issues

The figures presented in the manuscript have a very low quality.

The figures have been reinserted so they are more legible.